# The Effects of Association of Topical Polydatin Improves the Preemptive Systemic Treatment on EGFR Inhibitors Cutaneous Adverse Reactions

**DOI:** 10.3390/jcm10030466

**Published:** 2021-01-26

**Authors:** Mauro Bavetta, Dionisio Silvaggio, Elena Campione, Pietro Sollena, Vincenzo Formica, Deborah Coletta, Grazia Graziani, Maria Concetta Pucci Romano, Mario Roselli, Ketty Peris, Luca Bianchi

**Affiliations:** 1Dermatology Unit, Department of Systems Medicine, Tor Vergata University Hospital Foundation, 00133 Rome, Italy; campioneelena@hotmail.com (E.C.); luca.bianchi@uniroma2.it (L.B.); 2Dermatology Unit, Department of Medical and Surgical Sciences, Fondazione Agostino Gemelli University Hospital IRCCS, 00168 Rome, Italy; pietrosollena@gmail.com (P.S.); Ketty.Peris@unicatt.it (K.P.); 3Oncology Unit, Department of Systems Medicine, Tor Vergata University Hospital Foundation, 00133 Rome, Italy; v.formica1@gmail.com (V.F.); coletta.debbie@gmail.com (D.C.); mario.roselli@ptvonline.it (M.R.); 4Department of Systems Medicine, University of Rome Tor Vergata, Via Montpellier 1, 00133 Rome, Italy; graziani@uniroma2.it; 5Dermatology Unit, San Camillo Forlanini Hospital, 00152 Rome, Italy; pucciromano55@gmail.com; 6Institute of Dermatology, Cattolica del Sacro Cuore University, 00168 Rome, Italy

**Keywords:** EGFR inhibitors, polydatin, papulopustular rash, cutaneous adverse events

## Abstract

Epidermal Growth Factor Receptor inhibitors (EGFRi) are approved as therapeutic options in several solid tumors. Cutaneous papulopustular eruption is the most frequent cutaneous adverse-event (AE), usually treated with emollient or corticosteroids according to toxicity grade. Our study evaluated the efficacy and safety of a topical product containing polydatin, a glycosylated polyphenol, natural precursor of resveratrol showing anti-inflammatory and anti-oxidative activities, for the prevention and treatment of skin papulopustular rash in EGFRi-treated patients. Forty oncologic patients treated with EGFRi were enrolled in two groups: group-A, 20 patients with papulopustular AE, and group-B, 20 patients without cutaneous manifestations. The study consisted of twice-daily application of polydatin cream 1.5% (group-A) and 0.8% (group-B) for 6 months. In group-A patients, we observed at week 4 a remarkable improvement of skin manifestation and quality of life evaluated with National-Cancer-Institute-Common-Terminology-Criteria for Adverse-Events (NCI-CTCAE), Dermatology-Life-Quality-Index (DLQI) score and Visual-Analogue-Scale (VAS) pruritus, with a statistical significance of *p* < 0.05. None of the patients of group-B developed skin AEs to EGFRi. No cutaneous AEs related to the polydatin product were reported in both groups. Polydatin can be a good topical aid for the prevention and management of papulopustular rash in cancer patients receiving EGFRi, also capable of improving cancer patients’ quality of life.

## 1. Introduction

In recent decades, several new molecular target therapies have been developed and introduced for the treatment of cancers. Among these, important results have been achieved using Epidermal Growth Factor Receptor inhibitors (EGFRi) in different epithelial cancers. The EGFR or ErbB receptor belongs to type I tyrosine kinase (TK) membrane receptor family, which includes EGFR or ErbB1/Her1, ErbB2/Her2, ErbB3/Her3, and ErbB4/Her4. ErbB proteins act as homo- or heterodimers and contain an extracellular domain composed of four chains. Their role in the pathogenesis and development of epithelial malignancies is well known [1,2]. Activation of EGFR and its receptors’ family members results in receptor dimerization and tyrosine autophosphorylation, being crucial in the differentiation, proliferation and migration as well as adhesion, anti-apoptosis and survival of epithelial cells. EGFR and Her2 are frequently mutated or overexpressed in a variety of solid tumors [3]. Nowadays, EGFR inhibitors, including monoclonal antibodies and small-molecule TK inhibitors (TKIs), are approved as therapeutic options in solid tumors such as colorectal, head and neck, breast and lung cancers, since their overexpression or mutations are implicated in the pathogenesis of these cancer types [4]. As EGFR is mostly expressed on basal keratinocytes, its inhibition determines in these cells apoptotic effects, growth cell inhibition, higher adhesive cell capacity, minor cell migration, differentiation and inflammatory processes, through the release of inflammatory cytokines, leukocytes recruitment and release of pro-apoptotic molecules.

Among the possible adverse events (AEs), skin papulopustular rash is the most frequent during treatment with EGFRi (45–100%). Other reported skin AEs include dry skin, onycodystrophy, hair growth alteration, mucositis, and less frequently, hyperpigmentation and telangiectasia [4,5,6,7].

The typical papulopustular rash usually develops within 1–2 weeks from the beginning of therapy, with a peak at week 3; sometimes the onset of skin eruption may occur earlier, either in the second day of therapy, or later, after several months of therapy [8]. In addition, several clinical studies have shown that the occurrence of a severe anti-EGFR-induced skin AE correlates with better treatment response and longer survival [9,10,11]. A recent review by Hirayama et al. analyzed 25 studies evaluating the impact on Quality of Life (QoL) of skin toxicity (including skin rash, xerosis, pruritus, and paronychia) associated with EGFRi, being more pronounced in patients above 81 and those under 50 years of age [12].

To reduce EGFRi-induced skin AEs, several systemic drugs and topical preparations have been proposed, including tazarotene and adapalene but with no substantial relief of the skin rash [13]. Randomized clinical trials on the impact of minocycline as a pre-emptive therapy demonstrated a reduction in the incidence and severity of skin rash leading to its use in common clinical practice, which is currently recommended by international societies [14,15,16]. Since the use of tetracyclines only partially reduces the incidence of skin rash and as broad-spectrum antibiotics may markedly affect patient’s intestinal microbiota, new molecules are needed to ameliorate the discomfort of the patient and avoid stopping treatment or reduce the dose of EGFRi, with negative consequences on the primary disease [17].

Polydatin is a glycosylated polyphenol with a monocrystalline structure and it is a natural precursor of resveratrol. Firstly isolated from the plant *Polygonum cuspidatum*, polydatin is found in many products of daily diet, such as grape, peanuts, hop cones, red wines, hop pellets, cocoa-containing products, and chocolate products. It has anti-inflammatory, immunoregulatory, anti-oxidative and anti-tumoral proprieties. This glycosylated polyphenol mediates cell arrest and apoptosis, having a cytotoxic action on colorectal cancer cells [18,19,20]. The anti-inflammatory action is effective on keratinocytes, improving post-burning injury [21,22]. The modulation on inflammatory responses of human keratinocytes is due to the inhibition of the transcription factors nuclear factor kappa B (NF-κB), aryl hydrocarbon receptor (AhR) and epidermal growth factor receptor (EGFR)-extracellular-regulated kinase (ERK) pathways [23]. In addition, plant polyphenols, by acting on EGFR pathway, regulate chemokine expression and help tissue repair in human keratinocytes [24].

We have conducted a pilot study to evaluate the efficacy and safety of topical polydatin cream, on cutaneous AEs related to EGFRi.

## 2. Materials and Methods

We performed a prospective study including consecutive adult cancer patients in treatment with EGFRi for their tumors (colorectal, head and neck and lung cancers) followed and treated from September 2018 to October 2019 at the Oncology Unit of Policlinico Tor Vergata Rome.

The exclusion criteria were concomitant and/or pre-existing skin disease, breastfeeding or pregnant patients, ongoing topic or oral treatment with antioxidants or corticosteroids, poor patient compliance and previous treatment or well-known allergy/sensitivity to polydatin.

All enrolled patients were under anti-EGFR treatment regardless of the start of therapy, and they were divided in two groups: group A including patients who showed a skin rash at the first dermatologic examination and group B including patients without skin manifestations. 

During the study period, patients were examined every 4 weeks for the first 12 weeks (W0, W4, W8 and W12) and then after 12 more weeks (W24).

The primary endpoint of this study was to evaluate the efficacy of topical polydatin in patients who developed cutaneous papulopustular rash during treatment with EGFRi; secondary endpoint was to evaluate the possible preventive effect of topical polydatin on the onset of skin adverse events in symptoms-free patients.

For the overall evaluation of cutaneous and mucous adverse events, we used the National Cancer Institute Common Terminology Criteria for Adverse Events (NCI-CTCAE) grading scale, which recognizes 5 degrees of severity ranging from the lowest to death.

During each protocol visit, the patient had to fulfil the Dermatologic quality of life questionnaire (DLQI).

The same timing has been applied for the evaluation of pruritus, using the Visual Analogue Scale (VAS) score, ranging from 0 to 10, where 0 indicates no itch and 10 the most severe itch possible. 

The study protocol provided a twice daily topical application of a polydatin-based product, for at least twelve consecutive weeks. A cream or gel formulations were used based on the different site of application (skin, or mucous membrane). The following two formulations of the polydatin-based product have been applied: PH800 which has 0.8% of polydatin and PH151 with a concentration of 1.5%. Dermatological composition prepared according with European patent WO2019016843A1.

The PH151 formulation was assigned for the first 4 weeks to patients with skin papulo-pustular rash (group A). The PH800 formulation was prescribed to patients of group B since the first visit, and to patients of group A after the first 4 weeks, until the last visit of the study. All subjects were under long-term preventive treatment with minocycline 100 mg per day, as recommended by recent literature evidences and minocycline therapy was not discontinued during the study in accordance with the colleagues of the Oncology Unit and guidelines for the treatment of EGFRi skin AEs [13,25].

For statistical analysis, patients were grouped based on the presence or absence of cutaneous AE (group A and group B). Quantitative data (DLQI, NCI-CTCAE, VAS pruritus) were analyzed by Student’s t-test realized using Microsoft Office Excel. 

The study was approved by ethics committee at the participating center with the protocol number 146/2018 and was conducted in accordance with the Declaration of Helsinki and Good Clinical Practice guidelines. All patients signed the informed consent and were considered suitable to participate in the study.

## 3. Results

Forty patients (21 men and 19 women; mean age 68.23 years) treated with EGFRi (cetuximab, panitumumab, afatinib, gefitinib, osimertinib) were enrolled and divided in two groups.

Of these, 26/40 (65%) patients were suffering from metastatic colon cancer, 12/40 (30%) metastatic lung cancer, 1/40 (2.5%) metastatic laryngeal cancer and 1/40 (2.5%) metastatic rectal cancer.

The characteristics of group A and group B were a mean age of 65.84 years (range 48–83) and 70.63 years (range: 50–81) and a mean duration of disease of 3.32 years (range: 0.95–14.89) and 4.38 years (range: 0.7–14.91), respectively.

In the group A we observed an erythematous papulo-pustular skin rash in 20/20 patients (100%), periungual inflammation in 2/20 (10%) and presence of erythema and desquamation in 4/20 (20%) patients (Table 1).

We observed a statistical significance improvement (*p* < 0.05) of the cutaneous papulopustular rash at week 4 (W4), as shown in the trend of NCI-CTCAE chart, for all 20/20 patients using PH151. A remarkable improvement was observed in the DLQI score and pruritus VAS evaluation at W4 with a statistical significance of *p* < 0.05 (Figure 1, Figure 2, Figure 3 and Figure 4).

Patients from group B, who were symptoms-free at baseline and treated with PH800 (i.e., the polydatin-based product at the lower concentration), did not develop any skin adverse events, except for one patient who showed a mild erythema at W4 (NCI-CTCAE = 1). In this case, the topical preventive therapy was not modified and the patient reached complete clearance at week 8 (W8). The tables of quality of life (Figure 2 and VAS pruritus (Figure 3 reported a gradual improvement in values for group B. 

Recurrence of similar papulo-pustular skin rash was detected in 3 patients of group A during the maintenance period with PH800 cream; 1 patient at W8 and 2 patients at week 12 (W12). The new skin toxicities manifested as papulo-pustular rash, were evidenced by increase in NCI-CTCAE values (≥2) and worsening of DLQI (>10). The recurrences were successfully treated applying 1.5% polydatin cream as in the first 4 weeks of the study protocol for these patients belonging to group A. After 4 weeks of re-treatment, in all the 3 patients, there was a marked cutaneous improvement, confirmed by the decrease in the NCI-CTCAE and DLQI scores.

At the last observation, week 24 (W24), all patients from both groups were cutaneous disease-free, without any side effects related to the topical study products.

## 4. Discussion

The polydatin effect on the EGFR pathway of keratinocytes has been demonstrated by in vitro studies, as well as its action on reducing inflammation and promoting skin regenerative function [24,26].

Several pro-inflammatory cytokines, such as tumor necrosis factor alpha (TNFα) or interferon gamma (IFNγ), physical stimuli (UV irradiation, ionizing radiation, heat), cisplatin, and H_2_O_2_ can activate EGFR, interfering with cellular functions and causing the release of additional proinflammatory cytokines, through modulation of NF-κB and activator protein-1 (AP-1) transcription factors.

The cytoplasmic pathway of EGFR activates adaptive inflammatory/stress responses in keratinocytes, influencing their resistance to oxidative stress and heat shock [24]. Polydatin benefits have been largely attributed to its chain-breaking antioxidant or free radical scavenging activities. The clinical use of polydatin in skin disorders is also supported by the evidence that interleukin-8 (IL-8) plays a key role in inducing cutaneous AEs associated with anti-EGFR treatment, since polydatin is known to regulate IL-8 gene expression [22,27].

Ravagnan et al. demonstrated the capability of resveratrol and polydatin to regulate IL-6, IL-8 and TNF-alpha gene expression in heat-stressed human keratinocytes (HaCat). After pre-treatment with polydatin and resveratrol for 24 h, HaCat showed a reduced expression of these genes. Both molecules also contributed to skin regeneration and prevented cell damage, upregulating the heat shock protein 70B’(Hsp70B’) [20].

Two different in vitro studies evidenced the down-regulation of IL-6, IL-8 and TNF-alpha influenced by resveratrol and polydatin. [23,24]. Fuggetta et al. evaluated the effect of topical application of a moisturizer containing polydatin to prevent skin rash in 34 patients with mutated non-small cell lung cancer (NSCLC) treated with the EGFR-TKi afatinib. The results suggested that a polydatin-based cream can reduce the incidence of moderate to severe skin toxicities without additional side effects [28].

Our study showed a progressive clearing of the skin AEs secondary to different EGFRi therapies (cetuximab, panitumumab, afatinib, gefitinib, osimertinib) and an improved QoL post treatment with polydatin-based products. The topical treatment was extremely safe and no side effects were detected, confirming data from literature [28].

According to our data, the high compliance and patients’ satisfaction may be explained by the slight but constant improvement of the pruritus (VAS pruritus), probably due to xerosis reduction.

We reported a rapid improvement of the skin papulo-pustular rash in patients treated with the higher concentration of polydatin-based product (PH151), already noticed at the first control visit (W4) in 12/20 group A patients (Figure 2).

Only 3 patients of group A presented a worsening of the skin rash after a new cycle of oncological treatment (one patient at W8 and 2 patients at W12). In these cases, following the study protocol, patients re-applied the product with the higher concentration of polydatin (PH151) for 4 weeks, leading to resolution of the AE. Polydatin cream provided a quick and safe reduction or control of the common cutaneous adverse events related to antineoplastic EGFRi therapies to which oncologic patients must necessarily undergo. It could also act as a corticosteroid-sparing alternative due to its well-known anti-inflammatory and anti-oxidative properties.

## 5. Conclusions

The severity of cutaneous adverse events was reported to statistically correlate with the efficacy of EGFRi therapies and this could help clinicians in managing these skin rashes, although QoL is often decreased in functional evaluations [29].

Polydatin proves to be an excellent aid to prevent, attenuate or clear the skin papulopustular rash in cancer patients receiving anti-EGFR drugs. The improvement of these common drug-related skin rashes by applying polydatin-based products may also improve patient compliance to long-term cancer therapies, also improving patients’ QoL. We believe that a multidisciplinary approach, with a close cooperation between oncologists and dermatologists, could be the most effective way to manage the AEs related to EGFRi therapies.

Polydatin-based topical products should be counted as an effective choice for the treatment of skin papulopustular rash secondary to EGFRi therapy. Additional controlled studies, with a wider number of patients, are recommended to fully investigate its beneficial role.

## Figures and Tables

**Figure 1 jcm-10-00466-f001:**
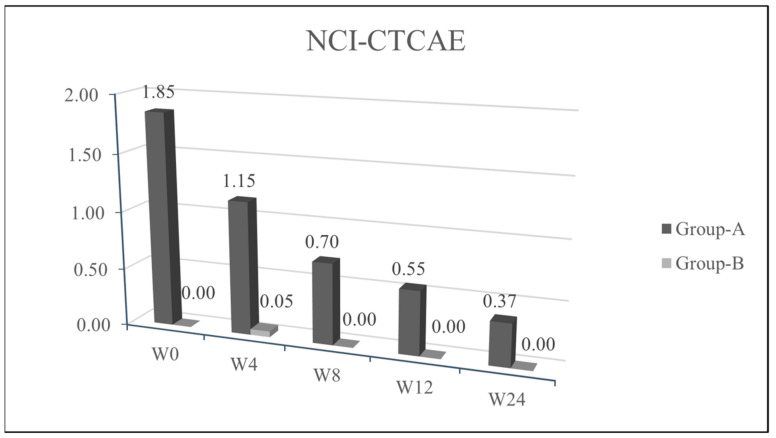
Average values of National Cancer Institute Common Terminology Criteria for Adverse Events (NCI-CTCAE) (*y*) in the various evaluation timings (*x*). *p* ≤ 0.05 at W4.

**Figure 2 jcm-10-00466-f002:**
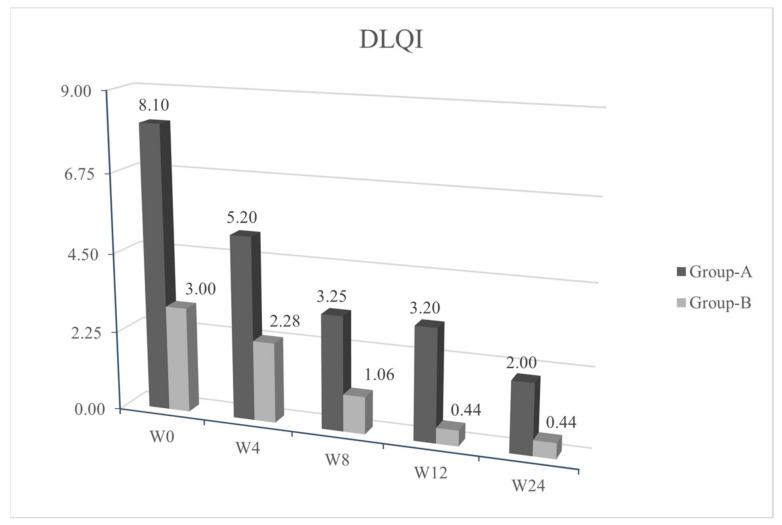
Average values of Dermatologic Quality of Life Questionnaire (DLQI) (*y*) in the various evaluation timings (*x*). *p* ≤ 0.05 at W4.

**Figure 3 jcm-10-00466-f003:**
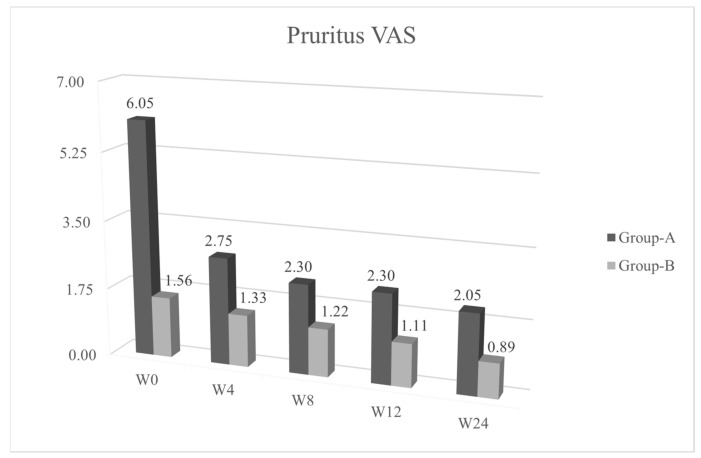
Average values of Visual Analogue Scale (Pruritus-VAS) (*y*) in the various evaluation timings (*x*). *p* ≤ 0.05 at W4.

**Figure 4 jcm-10-00466-f004:**
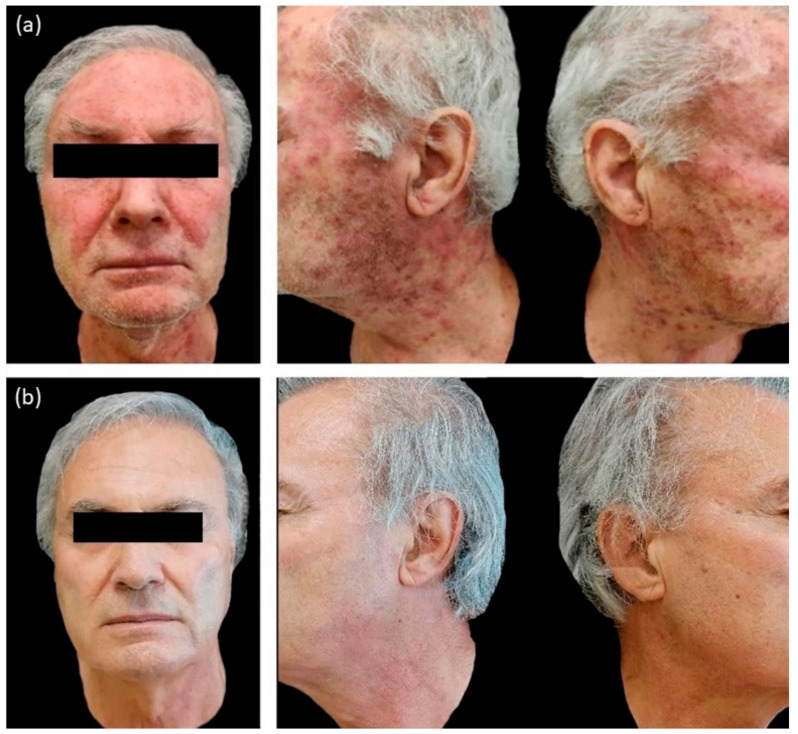
(**a**) Baseline pictures showing a papulopustular eruption of the face and neck due to Epidermal Growth Factor Receptor inhibitors (EGFR inhibitors). (**b**) Same patient after 4 weeks of treatment with polydatin 1.5% cream.

**Table 1 jcm-10-00466-t001:** Skin manifestations.

	AEs (rash)	
	N° of Patients	Percentage (%)
**Papulo-pustules**	19	98
**Periungual inflammaion**	2	10
**Erythema and Desquamation**	4	20

## Data Availability

Data available on request due to restrictions for privacy. The data presented in this study are available on request from the corresponding author.

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
