# Peer review of "The Effects of Association of Topical Polydatin Improves the Preemptive Systemic Treatment on EGFR Inhibitors Cutaneous Adverse Reactions"

_jcm, 2021, doi:10.3390/jcm10030466_

Round 1

Reviewer 1 Report

Dear authors.

I am very interstng about EGFR-related skin toxicity management.

I really thanks about your this clinical trial.

But, I would like to recommend some points for definite proof efficacy of topical polydatin.

Major concerns

  1. Number of enroled patient are not sufficent for identifing efficacy of investigational drug. I think at least 50 patients each group are needed for satisfy the statistical power even phase II study.
  2. Placebo or control arm are needed.
  3. if possible, tissue evaluation study could improve evidence of topical polydatin action mechanism.  

Author Response

Dear reviewer,

thanks for the time you have dedicated to our article and for your suggestions.

Regarding its major reviews;

We agree with your considerations, but our ethics committee approved the number of patients enrolled in our study (40 patients) because of the critical clinical characteristic of these subjects and it did not allow to perform histological evaluations. We considered the group B patients as possible comparison arm. We underline that our experience could be considered as a preliminary pilot observation that need further controlled studies including a wider number of patients to fully investigate the beneficial role of polydatin. Design and enroll additional controls or placebo groups in 5 days, as requested by the editor, it seems really hard to perform.

We look forward for further clarifications, in needed.

Thank you for the attention,

The authors.

Reviewer 2 Report

40 patients were included in this study 20 group A with AE B without AE

The results of this study are spectacular particularly for the highest dosage of polyadetin

It is not clear how the authors included the patients and why they are same number, did they recruit patients with a target number. It's not clear why there are the same number

The authors assess the preventive aspect of using polyadetin at low concentration while good therapeutic performance in group A (patients with lesion)

A best designed would have been to randomly assigned one or the other dosage to all patients under EGFRi the authors should discuss why they did not choose this option

There are no information about potential side effect of either dosage maybe there was none, this should be stated

Table 1 display 2 groups but doesnt provide comparison between both

Did the author discontinued the tetracyclin treatment? They could also discuss the option regarding their results to give more perspective

Author Response

Dear reviewer,

thanks for the time you have dedicated to our article and for your suggestions.

1) The study, the enrollment and group assignment was conducted according to the study protocol approved by our ethics committee. In particular we had a target number of 20 patients for each group assigned up the characteristic of having experienced or not the skin AE. 

The randomization could not be applied because in group A were included only patients with skin AE, while in group B patients not presenting AEs.

2) In our study no side effect or drug interactions from polydatin were reported, independently from the dosage used. This is particularly important in oncologic patients assuming multiple drugs (see pag 7 line 202).

3) Table 1 was designed to show the demographic characteristics and the heterogeneity of the two study groups, due to the representation difficulty, we decided to delete the table (see pag 3 table 1 line 129).

4) Minocycline therapy was not discontinued during the study in accordance with the colleagues of the Oncology Unit and guidelines for the treatment of EGFRi skin AEs. We modified the part of the manuscript in which the therapy with minocycline was mentioned to better explain our decision (see pag 3 line 117-119).

As we wrote in our article, additional controlled studies, with a wider number of patients, are recommended to fully investigate the beneficial role of polydatin, considering this as a pilot study adducing preliminary data of safety and efficacy.

We look forward for further clarifications, in needed.

Thank you for the attention,

The authors.

Round 2

Reviewer 2 Report

Answers have been provided in the cover letter

Author Response

Thanks for your attention and previous suggestions.

The Authors

This manuscript is a resubmission of an earlier submission. The following is a list of the peer review reports and author responses from that submission.